# FindThis: Language-Driven
# Object Disambiguation in Indoor Environments

**Arjun Majumdar**[2,*]    **Fei Xia**[1]    **Brian Ichter**[1]    **Dhruv Batra**[2]    **Leonidas Guibas**[1,3]

[1]Google DeepMind    [2]Georgia Institute of Technology    [3]Stanford University

[*] Work done while at Robotics at Google.

**Abstract:** Natural language is naturally ambiguous. In this work, we consider interactions between a user and a mobile service robot tasked with locating a desired object, specified by a language utterance. We present a task *FindThis*, which addresses the problem of how to disambiguate and locate the particular object instance through a dialog with the user. To approach this problem we propose an algorithm, *GoFind*, which exploits visual attributes of the object that may be intrinsic (e.g., color, shape), or extrinsic (e.g., location, relationships to other entities), expressed in an open vocabulary. *GoFind* leverages the visual common sense learned by large language models to enable fine-grained object localization and attribute differentiation in a zero-shot manner. We also provide a new visio-linguistic dataset, 3D Objects in Context (*3DOC*), for evaluating agents on this task, consisting of Google Scanned Objects placed in Habitat-Matterport 3D scenes. Finally, we validate our approach on a real robot operating in an unstructured physical office environment using complex fine-grained language instructions.

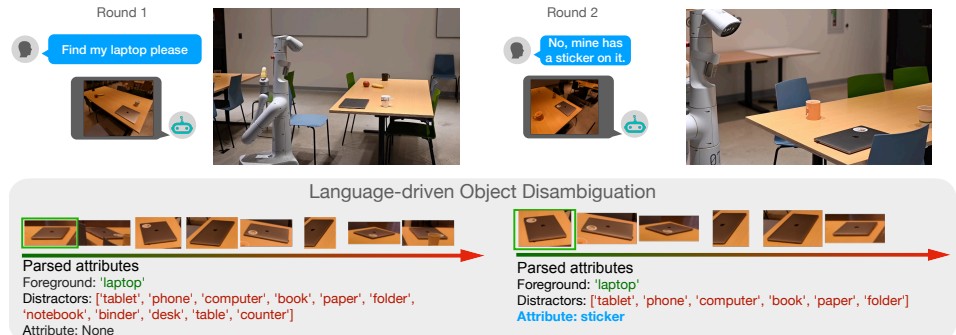

Figure 1: We propose *FindThis*: an object navigation task in which an agent engages in multiple rounds of open-vocabulary, natural language interaction to locate a specific object in a complex environment. We present *GoFind*, an approach to parse the instruction and robustly find objects by identifying attributes and removing distractors. In this example, though the robot initially finds a laptop, the user is able to provide additional attributes ('stickers on it') to identify the correct object.

## 1   Introduction

A long-standing goal in robotics is to develop agents that can follow natural language instructions to find objects in the real world. From the onset, researchers have envisioned systems capable of identifying specific objects through a free-flowing dialog – allowing users to specify and clarify instructions (e.g., SHRDLU [1]). Recently, progress has been made by on sub-tasks within this grand challenge. For example, one line of research has centered on grounding referential language (e.g., *'Find the lamp next to the sofa.'*) in fully-observable 2D images [2] or 3D scenes [3, 4, 5]. Complementary work has focused on developing agents that can visually navigate (i.e., take actions) through partially-observable 3D environments by following step-by-step navigation instructions (e.g., *'Exit the bedroom. Walk down the hall. Stop next to the bathroom sink.'*) [6, 7, 8]. Furthermore, researchers have developed agents that explore 3D environments to find objects given concise instructions that specify objects by category labels (e.g., *'Find a table'*, *'Find a bed'*, *'Find a chair'*, etc.) [9, 10].

7th Conference on Robot Learning (CoRL 2023), Atlanta, USA.

Given the tremendous progress in each of these domains, we believe it is time to ask: what type of user interaction should ultimately be supported by robots tasked with finding objects? Towards answering this question we identify three key desiderata:

**1) Specificity.** Users should be able to request a specific instance of an object. For example, when asking for a *'laptop'* we often would like **our** *'laptop'* and not one belonging to someone else. Thus, as illustrated in Figure 1, users might specify distinguishing features such as *'the laptop with stickers on it'* or *'the laptop on the bed'*. This requires systems to differentiate intrinsic attributes (e.g., color, patterns, shape) and/or extrinsic properties (e.g., spatial location, relationship to other entities).

**2) Open-vocabulary.** The set of objects should not be restricted – users should be able to request any object they require. Thus, systems must support an open vocabulary [11, 12, 13].

**3) Interactive.** When providing instructions users may make ambiguous or even mistaken requests. For example, a user might request an *'orange mug'* when multiple mugs of that color exist. In such cases, the request is underspecified and further interaction is required to provide additional details or corrections.

To evaluate agents under these real-world conditions, we introduce a task and dataset. As shown in Figure 1, our task (*FindThis*) requires following natural language instructions (e.g., *'Find my laptop.'*) to localize a specific object in an indoor 3D environment such as a home or office. If the agent fails by finding an incorrect object (as in Figure 1), the task continues with the user providing further details about the desired object (e.g., *'No, mine has a sticker on it.'*). In other words, *FindThis* agents must engage with users in a multi-round interaction, interpreting open-vocabulary, natural language instructions to localize a desired object.

In addition, we present the 3D Objects in Context (*3DOC*) dataset, which is designed for large-scale evaluation of *FindThis* agents in a zero-shot manner – i.e., the *3DOC* dataset does not include a training split. The dataset consists of language instructions generated from dialog templates, paired with scenes constructed using a diverse set of 3D objects from the Google Scanned Objects (GSO) dataset [14] and placed in 3D scanned indoor environments from the Habitat-Matterport 3D (HM3D) dataset [15].

To address the *FindThis* task, we propose a novel zero-shot approach (*GoFind*) to handle the challenges of multi-round interaction for fine-grained object localization. In our approach, we build an open-vocabulary 3D scene representation using vision-and-language models (e.g., ViLD [16], OWL-ViT [17], CLIP [18]). Then, we query the representation with the help of a large language model (e.g., GPT-3 [19] or PaLM [20]), which identifies object attributes mentioned in the dialog and uses visual common sense (i.e., knowledge about visual similarity) to propose background objects that the embodied agent (i.e., robot) should ignore.

Finally, we validate our approach on both *3DOC* and on a real robot operating in a real-world office environment using complex fine-grained instructions such as *'Find a bowl full of cereal'* or *'Find an upside down mug'*. We find that our proposed approach generalizes well to the real-world setting, performing as well or better than the performance observed in simulation.

## 2    Related Work

**2D and 3D Referring Expression Tasks.** Aligning referential language (e.g., *'the mug on the table'*) with image or spatial regions has been studied in both 2D [2] and 3D [3, 4, 5] settings. A common framing of this task is a game between two players in which both players fully observe a scene, then one describes an object that the second player must identify [2]. The 3D versions of this game (e.g., ReferIt3D [3], ScanRefer [4], Refer360 [5], REVERIE [21], SOON [22]) require fine-grained disambiguation of objects based on intrinsic and extrinsic attributes. Our proposed task (*FindThis*) expands on this prior work by lifting this problem into a real-world setting in which target objects are described through a multi-round dialog with a user and an embodied agent must explore 3D environments, observing objects at different distances and viewing angles and ignoring distractor objects that share some but not all properties with the target.

**Language-Conditioned Visual Navigation Tasks.** Language-conditioned navigation in 3D environments is often studied through instruction following [23, 24, 8, 25, 26, 27] or object-goal navigation (ObjectNav) task [9, 10]. This work falls into ObjectNav, which prior work has focused on a setting in which objects are only described by a category label (e.g., *'sink'*, *'table'*, etc.) drawn from a closed set of categories, and finding any instance of an object with that label is defined as success[10, 28, 29, 30, 31, 32, 33]. By contrast, we study an alternative setting in which the user prefers a specific instance of an object (e.g., *'the orange mug'*) drawn from an open set of categories that are not predetermined. Recent work has studied a similar open-vocabulary setting [11, 12, 13]. Closely related, [12] propose an open-vocabulary object

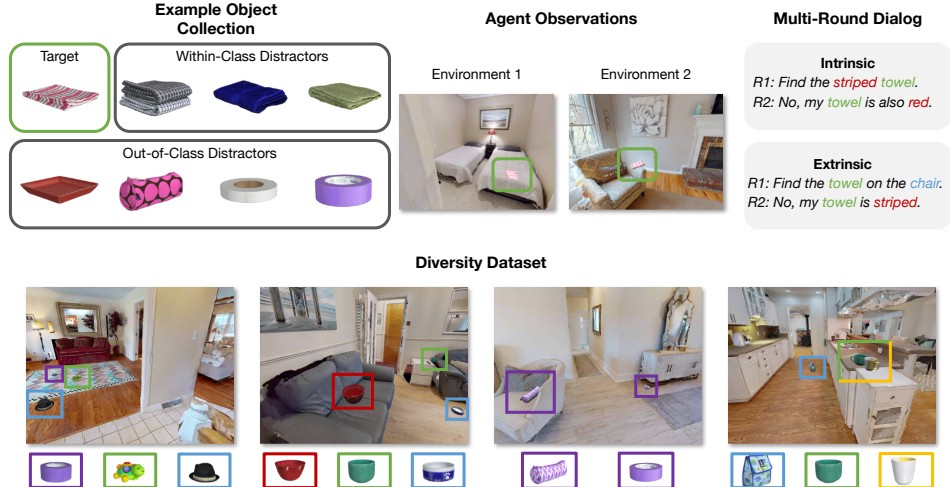

Figure 2: Examples from the *3DOC* dataset. *3DOC* is designed to benchmark the ability of agents to parse multi-round *open-vocabulary* dialog to identify objects in a 3D scene. Compared with other object-in-scene datasets, *3DOC* is designed to include more challenging within-class and out-of-class distractors and includes more diversity in objects and scenes.

navigation task coined *language-driven zero-shot object navigation* (L-ZSON). The primary difference between L-ZSON and the task proposed in this work (*FindThis*) is that we consider a multi-round dialog, which is often required for disambiguating similar object instances.

**3D Semantic Mapping.** Numerous methods have been proposed for building semantic maps of 3D environments. One line of work adds semantic features to traditional SLAM methods. Alternatively, several works propose using object detectors or semantic segmentation techniques to build semantic maps. A key limitation of these methods is that the detectors and segmentors are pre-trained on a fixed set of object categories. This makes such methods inapplicable to the open-vocabulary setting studied in this work.

Recent methods have addressed this challenge using recently proposed open-vocabulary vision-and-language models (e.g., CLIP [18], ALIGN [34], BASIC [35]) and object detectors and segmentors (e.g., ViLD [16], MDETR [36], OWL-ViT [17], LSeg [37]). Specifically, NLMap [38] uses a ViLD detector and CLIP features to represent 3D environments. VLMaps [39] leverages LSeg [37], a model that projects 2D images into the CLIP representation space at the pixel-level, for mapping. And the best version of CoW [12] uses the OWL-ViT detector. In all of these methods, objects are found using queries generated by a pre-trained text encoder. By contrast, in this work we consider more complex query mechanisms required to address the unique challenges in the *FindThis* task. Furthermore, we leverage a large language model (LLM) to provide common sense visual knowledge and decompose language instructions, improving object localization performance.

## 3 The *FindThis* Task and the *3DOC* Dataset

Our goal is to design a task in which a user *interacts* with an embodied agent using natural language with an *open-vocabulary* to describe a *specific* object in a 3D scene for the agent to find. In this section, we define such a task (Section 3.1), evaluation metrics (Section 3.2), and present a dataset (Section 3.3) designed for large-scale evaluations of embodied agents on this task in a zero-shot manner (Figure 2).

### 3.1 Task Definition

In **FindThis**, for each episode $i$ an embodied agent (a mobile robot) must localize a target object $T_i$ (e.g., '*a mug*') in a 3D environment $E_i$ through a multi-round dialog $D_i$ in which a user describes the object. Specifically, the agent is initialized at a starting location $s_0$ in the environment $E_i$, and must explore to find the target object $T_i$. In other words, given descriptions of an object, the agent must '*go find this*'.

By nature, language instructions can be ambiguous and underspecify the task. For example, a user might ask for their specific '*mug*' when multiple mugs are present in the scene. In such cases, users must provide additional information to resolve ambiguities. Specifically, the additional information may describe

**intrinsic attributes** of the object (e.g., *'Find my orange mug'*) or **extrinsic attributes** such as the spatial relationship with another object in the scene (e.g., *'Find the mug next to the sink'*). A *FindThis* dialog may include both types of disambiguating information.

As illustrated in Figure 1, we consider multimodal dialog in which users provide natural language instructions such as *'Find my laptop please'* or *'No, mine has a sticker on it'*, and the agent responds with an image of a candidate object $C_j$ (e.g., an image of a laptop). Formally, a sequence of these user instructions $I_j$ and agent responses $C_j$ compose a $M$-round dialog $D_i = [I_1, C_1, ..., I_M, C_M]$. In this work, we consider $M \in \{1, 2\}$. An episode is considered successful if the candidate object matches the target (i.e., $T_i$ is in $C_j$).

## 3.2 Evaluation Metrics

We use two variations of success rate (SR) to measure performance: Top-1 SR and Top-5 SR. Top-1 SR (a standard variant) corresponds with the percentage of episodes in which the agent presents the user with the correct candidate object (i.e., $T_i$ is in $C_j$) within $M$ rounds of dialog. Top-5 SR considers a setting in which the agent can present five candidates in each round. If the target object is within the set of candidates, the episode is considered a success in terms of Top-5 SR. A target $T_i$ is considered within the image crop $C_j$ if the ground truth segmentation mask of $T_i$ covers at least 10% of the cropped image $C_j$.

## 3.3 Evaluation Dataset

This section describes the **3D Objects in Context (*3DOC*)** dataset, designed for evaluating agents on the *FindThis* task. *3DOC* is designed for zero-shot evaluation, so training episodes are not provided. We instantiate the task in the Habitat [40, 41] simulator by placing 3D scanned objects from the Google Scanned Objects (GSO) dataset [14] into photorealistic 3D environments from the Habitat-Matterport 3D (HM3D) dataset [15]. Examples are shown in Figure 2 and statistics summarized in Table 2 in Appendix B.

**Scene Construction.** We create scenes by procedurally dropping the 3D objects on surfaces within the HM3D [15] environments. Specifically, for each object we annotate if the object typically appears on elevated surfaces (e.g., a table, desk, counter, etc.), on the floor, or on both surface types. Then, we place the object above an appropriate surface location, drop the object using the simulator's physics engine, and verify that the object stably landed in close proximity (i.e., within 0.05m) of the desired surface location. This procedure ensures that objects are placed in semantically reasonable locations that might represent a messy house (e.g., with toys randomly placed on the floor). A similar approach for determining reasonable object placements was used to construct an evaluation dataset for an open-vocabulary object navigation task in [42].

A key challenge in *FindThis* is disambiguating *distractor* objects that are similar to the target object in some way. To include such scenarios in *3DOC*, we annotate the object category and intrinsic attributes for 72 objects from the GSO dataset [14]. We use these annotations to curate 50 object collections composed of two different types of distractors. Specifically, we sample up to 4 objects from a target object category with different intrinsic attributes and up to 4 objects from other categories that may share intrinsic attributes with the target objects. Thus, for a given target object, agents must disambiguate both within-class and outside-of-class distractors to solve a *FindThis* episode. An example object collection is shown in Figure 2. Finally, we place objects from each of the 50 object collections into 10 different HM3D [15] validation environments, which results in 500 unique scene layouts in 100 different HM3D environments.

**Intrinsic Attribute Instructions.** Given a target object from an object collection, we create an initial intrinsic attribute instruction using the template: `Find the <attr 1> <object>`, where `<attr 1>` corresponds to the most prominent intrinsic attribute of an object (e.g., its color or pattern) and `<object>` is a name for the object (e.g., *'mug'* or *'coffee mug'*). In subsequent rounds of interaction, a user response is synthesized using secondary intrinsic attributes (`<attr 2>`) with the template: `No, my <object> is also <attr 2>`. In total, we generate 1,713 unique episodes, for 10 different object categories and 53 unique attribute mentions to form the intrinsic attribute split of the *3DOC* dataset.

**Extrinsic Attribute Instructions.** Initial extrinsic attribute instructions are generated with the template: `Find the <object> <relation> <ref>`, where `<object>` is the target, `<ref>` is an object in proximity of the target, and `<relation>` is drawn from the set: {`on the, in the, next to the, near the`}. An example extrinsic attribute instruction is shown in Figure 2. In multi-round dialog for extrinsic attribute episodes, subsequent rounds use intrinsic attributes for further differentiation with the template: `No, my <object> is <attr>`, where `<attr>` is an attribute of the target. In total, the extrinsic attribute split of *3DOC* contains 50 unique episodes.

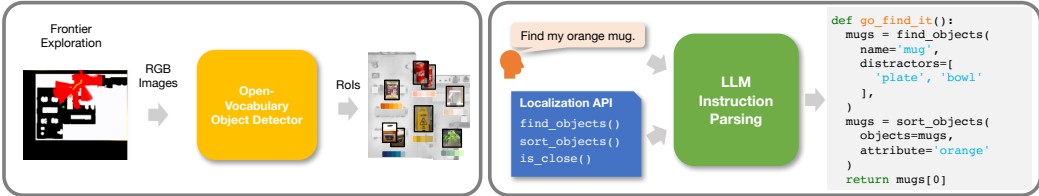

Figure 3: Overview of the *GoFind* algorithm. First, we use frontier exploration and class-agnostic object detection to obtain RoIs. CLIP [18] and ViLD [16] features are extracted from each RoI to establish an open-vocabulary 3D scene representation. Given a natural language query, we use an LLM to parse the target ('mug'), propose distractors (e.g., 'plate', 'bowl', etc.), and call functions from the localization API (operating on the open-vocabulary 3D scene representation) to locate the object.

Note, instruction templates are only used for generating a large-scale evaluation dataset, while our approach can handle open-vocabulary user interactions as demonstrated in real-world experiments (Section 5.3).

## 4 Approach

This section describes our approach (Figure 3) for multi-round, fine-grained object localization. In our approach, an agent first explores new environments to build a 3D semantic representation of the scene (Section 4.1). Then, an LLM with access to a fine-grained object localization API (Section 4.2) parses user interactions to propose objects to the user in a multi-round dialog (Section 4.3).

### 4.1 Open-Vocabulary 3D Scene Representation

As illustrated in Figure 3, our agent first explores the environment using frontier exploration [43]. At each timestep $t$, RGB observations are processed with an open-vocabulary object detector (ViLD [16] or OWL-ViT [17]) to produce regions-of-interest (RoIs). For each RoI, we calculate a 3D bounding box by projecting pixel locations corresponding to the object's predicted 2D segmentation mask (ViLD) or 2D bounding box (OWL-ViT) into 3D space using the depth observation and camera matrices $\mathcal{R}$. Then, we use the minimum and maximum projected 3D locations to define an RoI's 3D bounding box. For each RoI, we compute a CLIP representation using the CLIP visual encoder $CLIP_v$, which is used in addition to the RoI features produced by the object detector. As a result, the 3D environment is represented as a list of RoIs, where each RoI is represented by a 3D bounding box and 2 types of semantic features (CLIP plus ViLD or OWL-ViT). Note: because the object detector is run independently at each timestep, multiple RoIs might correspond to different views of the same object.

### 4.2 Fine-Grained Object Localization API

To perform fine-grained object localization on a 3D semantic scene representation, we need a mechanism to answer three basic questions: (1) Does object `X` exist? (2) Does object `X` have the intrinsic attribute `Y`? and (3) Does object `X` have an extrinsic relationship with another object `Z`? In this work, we design simple API calls that operate on the open-vocabulary scene representation (Section 4.1) to answer each question. Specifically, we frame the questions as a ranking problem where the goal is to sort the list of RoIs. Then, we design algorithms to answer each question that assign each RoI a score between 0 and 1. In Section 4.3, we demonstrate how a LLM can compose these API calls in response to multi-round *FindThis* user queries.

**1. `find_objects(obj, distractors)`** Given an object name `obj` (e.g., *'mug'*), our goal is to filter RoIs that do not contain an instance of the requested object, and then sort the remaining RoIs based on a confidence score. Open-vocabulary object detectors (e.g., ViLD [16] and OWL-ViT [17]), score RoIs by encoding object names `obj` with a text encoder ($CLIP_t$) and calculating a similarity score with the visual features for each RoI. These scores can be used to sort the RoIs and filter ones with scores below a threshold $\tau$. In this work, we dynamically calculate $\tau$ for each RoI based on the similarity score with a set of background categories. If an RoI is more similar to a background category than the desired object `obj` it is discarded. We generate background categories on-the-fly by asking an LLM to answer the question, what objects look similar to desired foreground object `obj`?

**2. `sort_objects(attribute)`** When an intrinsic attribute `attr` is provided, the list of RoI must be sorted according to both the object category `obj` and attribute `attr` to return an object that

| # | method | M=1 rounds of interaction | | | | | | M=2 rounds of interaction | | | | | |
|---|---|---|---|---|---|---|---|---|---|---|---|---|---|
| | | intrinsic | | extrinsic | | average | | intrinsic + intrinsic | | extrinsic + intrinsic | | average | |
| | | top-1 | top-5 | top-1 | top-5 | top-1 | top-5 | top-1 | top-5 | top-1 | top-5 | top-1 | top-5 |
| 1 | ViLD (no parsing) [38] | 16.7 | 28.8 | 10.5 | 26.3 | 13.6 | 27.6 | 23.9 | 37.6 | 28.9 | 44.7 | 26.4 | 41.2 |
| 2 | ViLD + basic parsing | 23.8 | 37.3 | 18.4 | 26.3 | 21.1 | 31.8 | 29.6 | 43.9 | 23.7 | 44.7 | 26.6 | 44.3 |
| 3 | ViLD + ours | 27.8 | 40.7 | 21.1 | 31.6 | 24.5 | 36.2 | 35.0 | 47.2 | 31.6 | 39.5 | 33.3 | 43.4 |
| 4 | ViLD+CLIP (no parsing) [38] | 33.4 | 44.3 | 7.9 | 21.1 | 20.6 | 32.7 | 41.2 | 50.4 | 36.8 | 52.6 | 39.0 | 51.5 |
| 5 | ViLD+CLIP + basic parsing | 38.7 | 48.5 | 13.2 | 28.9 | 26.0 | 38.7 | 45.0 | **53.3** | 31.6 | 42.1 | 38.3 | 47.7 |
| 6 | ViLD+CLIP + ours | **40.3** | **48.6** | **47.4** | **55.3** | **43.8** | **52.0** | **46.4** | 52.7 | **71.1** | **73.7** | **58.8** | **63.2** |
| 7 | OWL-ViT (no parsing) [12]* | 27.6 | 38.3 | 21.1 | 34.2 | 24.4 | 36.3 | 32.2 | 44.2 | 26.3 | 44.7 | 29.2 | 44.5 |
| 8 | OWL-ViT + basic parsing | 29.2 | 41.3 | 18.4 | 28.9 | 23.8 | 35.1 | 35.0 | 46.2 | 36.8 | 60.5 | 35.9 | 53.4 |
| 9 | OWL-ViT + ours | 30.2 | 42.2 | 36.8 | 44.7 | 33.5 | 43.5 | 34.8 | 46.4 | 57.9 | 63.2 | 46.4 | 54.8 |

Table 1: Main Results. We show results on the 3DOC benchmark by applying our proposed approach on top of three different semantic scene representations: ViLD, ViLD+CLIP, and OWL-ViT. We find our method with ViLD + CLIP performs the best. Both parsing and delta vector contributes to improvement of the performance. (*our implementation of CoW [12] in a pre-explored setting instead of object navigation in novel environment)

meets both criteria. To facilitate this, we propose using attribute *delta vectors* to generate scores for intrinsic attribute queries. Specifically, we define the *delta vector* as the difference in the CLIP text embedding space between the attribute conditioned object category (attr+obj) and the generic object category (obj):

$$\Delta(\texttt{attr},\texttt{obj}) = CLIP_t(\texttt{attr+obj}) - CLIP_t(\texttt{obj}) \tag{1}$$

Then, we use the magnitude of the projection of the visual features for each RoI onto the delta vector as an attribute score. Finally, we add the attribute score to the original object score produced by the find_objects() method to calculate a score for each RoI.

**3. is_close(obj1, obj2)** Given an language query expressing an extrinsic relationship between objects (e.g., *'Find my mug next to the sink'*) a minimum requirement is the two objects (*'mug'* and *'sink'*) are in close proximity. We mechanize this requirement by comparing the minimum distance between a pair of 3D bounding boxes with a fixed threshold (0.5m in our experiments). Future work may explore additional API calls that capture more refined spatial relationships such as 'left of' or 'behind'.

### 4.3 Instruction Parsing

A key challenge in the *FindThis* task is parsing the multi-round dialog with a user. While the *3DOC* uses templated language, a *FindThis* system should be able to handle arbitrary inputs, reflecting the open-vocabulary motivation for the task. As shown in Figure 3, in this work, we use a LLM with few shot prompting to translate multi-round user interactions into a go_find_it() function that uses methods from the localization API (Section 4.2) to return a candidate object $C_j$ to the user. Specifically, we use a small number of example dialog-to-function translations in a few-shot prompt to enable this capability. Further details including the prompt used in our experiments is presented in Appendix A.

## 5 Experiments

In this section, we evaluate our approach in both simulation using our 3DOC dataset (Section 5.2) and on a robot in a real-world kitchen in an office environment (Section 5.3).

### 5.1 Experimental Setup

**Occupancy Map.** Our experiments assume that agents are provided with a top-down 2D occupancy map that indicates navigable regions in the environment (Fig. 3). In simulation, the occupancy map is directly generated by the Habitat simulator. However, in our real-world experiments only a coarse occupancy map (similar to an empty floor plan) is provided. Specifically, the real-world occupancy map only delineates walls and other immovable structures (e.g., a kitchen island) but does not include movable objects such as tables or chairs. The robot uses RGB-D and lidar sensors to update the occupancy map in real-time, which allows navigating to waypoints without collisions.

**Exploration Strategy.** For all experiments, all agents (including baselines) use the occupancy map to pre-explore the environment using frontier exploration [43]. Specifically, at each timestep, visible regions

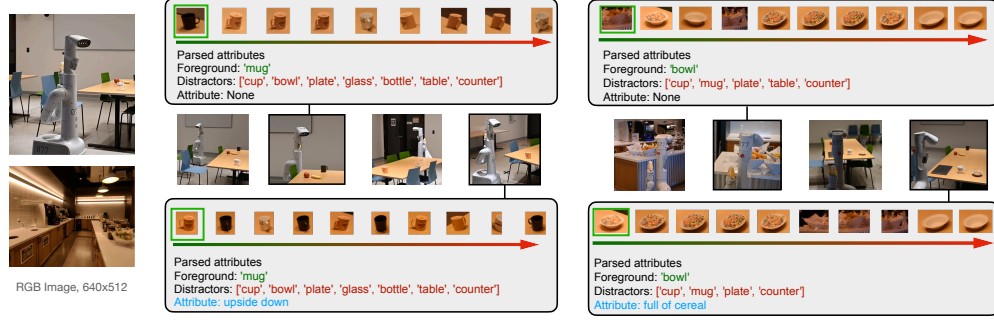


RGB Image, 640x512

Robot setup    Instruction: Find a mug -> Find an upside down mug    Instruction: Find a bowl -> Find a bowl full of cereal


Figure 4: **Real robot qualitative results.** We implemented our approach on a mobile robot and show qualitative results in which our algorithm guides the robot through the environment to find objects specified by complex fine-grained language, e.g., 'an upside down mug' or 'a bowl full of cereal'.

within a fixed radius of the agent are marked as 'explored.' Then, the agent selects the nearest waypoint along the boundary (or frontier) of the explored area to navigate to next. The agent updates the explored regions along the way and stops exploring when all of the navigable regions have been marked 'explored.'

**Baselines.** We implement variations of our method to compare with two state-of-the-art methods from prior work: NLMap [38] and CoW [12]. Both methods use frontier exploration [43], an open-vocabulary object detector (e.g., ViLD [16] or OWL-ViT [17]) to build a scene representation, and a frozen text encoder to query the representation.

- **NLMap** [38] represents the scene as a list of RoIs produced by ViLD's class-agnostic region proposal network. Each RoI is represented with positional information (i.e., object location and size) and visual features from ViLD [16] and CLIP [18]. A frozen CLIP text encoder is used to query the representation.
- **CoW** [12] uses the OWL-ViT [17] detector to process images at each timestep during exploration. If the similarity score between a detected region and the query text exceeds a threshold, the CoW agent navigates to the 3D location corresponding with that RoI.

These methods as well as other prior works (e.g., VLMaps [39]) use a simple mechanism to query scene representations: cosine similarity between image and text features from a vision-and-language model. Specifically, language queries (e.g., *'Find the orange mug.'*) are feed directly to a CLIP text encoder – potentially, after some basic parsing such as removing the phrase *'Find the'*. By contrast, our approach uses an LLM to parse instructions into parts (e.g., *'orange'* and *'mug'*) that are used to query for objects.

To compare with these works, we implement two variations of our method: 1) **no parsing** where the full multi-round dialog is used as a text query and 2) **basic parsing** where the dialog is converted into a concise attribute-object phrase (e.g., *'orange mug'*) or referring expression (e.g., *'mug next the sink'*). Furthermore, we experiment with creating scene representations using three different feature sets: ViLD [16] features, ViLD [16] and CLIP [18] features (à la NLMap [38]), and OWL-ViT features (à la CoW [12]).

**Implementation Details.** We conduct experiments in simulation on the 1,763 total (1,713 intrinsic and 50 extrinsic) evaluation episodes in the *3DOC* dataset. We use the `code-davinci-002` version GPT-3 [19] for experiments in simulation and a 540B parameter model from PaLM for real-world experiments.

**Agent Embodiment.** We simulate an agent with an embodiment similar to a mobile manipulation robot produced by Everyday Robots (shown in Figure 1). Specifically, the agent has a height of 1.25m with an RGB-D camera with a 90° horizontal field-of-view (FOV) and a sensor resolution of 640×480. The camera is placed at the top of the agent. We assume the agent has access to an occupancy map indicating navigable areas in the environment. In real-world settings, such occupancy maps can be pre-collected and then updated on-the-fly using depth and/or lidar sensors. We simulate a discrete action space in which the agent can move forward 0.25m and turn left or right by 10°, where the turn actions are in-place rotations of the agent's base.

## 5.2 Results on *3DOC*

Table 1 shows results on the 3DOC benchmark by applying our proposed approach on top of three different semantic scene representations: ViLD, ViLD+CLIP, and OWL-ViT. We find that in all cases our approach

(rows 3, 6, and 9) improves top-1 success rate (SR) for both a single round ($k=1$) and two rounds of interaction ($k=2$). Specifically, with ViLD+CLIP our approach improves average top-1 success rate (SR) by **+17.8** points ($26.0 \rightarrow 43.8$) for $k=1$ over basic parsing (row 5 vs. 6). With two rounds of interaction ($k=2$), we observe similar gains in average top-1 SR of **+20.5** points ($38.3 \rightarrow 58.8$ from row 5 to 6).

For intrinsic attribute differentiation, we find that basic instruction parsing (e.g., converting the multi-round dialog *'Find my wicker basket. No, my basket is also white.'* to *'white and wicker basket'*) consistently leads to improvements (row 1 vs. 2, 4 vs. 5, and 7 vs. 8). For example, with the ViLD+CLIP representation basic parsing improves intrinsic top-1 SR by **+5.3** points ($33.4 \rightarrow 38.7$) for $k=1$ and **+3.8** points ($41.2 \rightarrow 45.0$) for $k=2$ (row 5 vs. 6). These results indicate that these scene representations are sensitive to spurious words unrelated to attributes or objects, and highlight the need for instruction parsing for multi-round interaction.

Furthermore, we find that our approach of using CLIP delta vectors for intrinsic attribute differentiation consistently improves top-1 SR (row 2 vs. 3, 5 vs. 6, and 8 vs. 9). For example, with the ViLD+CLIP representation intrinsic top-1 SR improves by **+1.6** points for $k=1$ ($38.7 \rightarrow 40.3$) and **+1.4** points $k=2$ ($45.0 \rightarrow 46.4$) (row 5 vs. 6). With ViLD representations, we observe larger improvements in top-1 SR of **+4.0** points ($23.8 \rightarrow 27.8$) for $k=1$ and **+5.4** points ($29.6 \rightarrow 35.0$) for $k=2$ (row 2 vs. 3). We emphasize that these scene representations are built on CLIP [18], which was trained to differentiate intrinsic attribute mentions that are likely contained in the WIT dataset [18]. Despite such training, our proposed approach substantially improves over this strong baseline.

For extrinsic differentiation, we find that our proposed approach of separating instructions (e.g., *'Find the mug on the table.'*) into its component parts and enforcing spatial constraints (i.e., proximity) significantly improves performance. For instance, with ViLD+CLIP representations, extrinsic top-1 SR improves by **+34.2** points ($13.2 \rightarrow 47.4$) for $k=1$ (row 5 vs. 6). We observe further gains for multi-round dialogs ($k=2$) when an intrinsic attribute is provided as an additional clue (e.g., *'Find the mug on the table. No, my mug is white and yellow.'*) of **+39.5** points ($31.6 \rightarrow 71.1$). These results highlight the benefits of composing multiple capabilities (intrinsic and extrinsic) through the API introduce in Section 4.2 that build on top of open-vocabulary scene representations for fine-grained object localization.

### 5.3 Qualitative Results on a Real Robot

As a last step, we deployed our method on a mobile robot to qualitatively show generalization to a real world setting. As seen in Fig 4, the robot is a mobile manipulator, deployed in an office kitchen environment, with a $640 \times 512$ RGB camera image observation. In each of our runs, the robot successful identifies a 'black upside-down mug' and a 'bowl full of cereal' through two rounds of interaction. Specifically, in the first round the robot find a different 'mug' or 'bowl', but correctly identifies the target after a second round of interaction. Owing to the scale and generality of the CLIP-based vision backbone, we find this real world application is high performing with no adaptions to the algorithm. See additional details in Appendix C.

## 6 Discussion

We proposed a novel algorithm called *GoFind* for fine-grained object localization in complex indoor environments using natural language queries and visual attributes. We addressed the challenge of disambiguating and locating the particular object instance desired through a dialog with the user, and exploited visual attributes of the object that may be intrinsic or extrinsic, expressed in an open vocabulary. We demonstrated that the visual common sense learned by large language models enables fine-grained object localization and attribute differentiation in a zero-shot manner. We also provided a new visio-linguistic dataset, 3D Objects in Context (*3DOC*), for evaluating agents on the *FindThis* task. Finally, we validated our approach on a real robot operating in an unstructured physical office environment using complex fine-grained language instructions.

**Limitations.** Our proposed method showed improvements in performance over existing methods on benchmark datasets and demonstrated qualitative results on a mobile robot. However, there are still limitations to our approach. A key limitation is that we use a pipeline of region proposal detection and feature extraction, which might miss some objects, causing irreversible misdetections. Another limitation is that we use frontier exploration to do exhaustive search in all our experiments, while some dynamic stopping algorithms could potentially provide a shorter path length. We hope that our work provide a solid baseline and a dataset for further research in this area and lead to the development of more robust and efficient algorithms for natural language object disambiguation in complex indoor environments.

## Acknowledgments

The Georgia Tech effort was supported in part by ONR YIP and ARO PECASE. The views and conclusions contained herein are those of the authors and should not be interpreted as necessarily representing the official policies or endorsements, either expressed or implied, of the U.S. Government, or any sponsor.

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

# A    Few-Shot Prompt

We use the few-shot prompt shown in Listing 1 for the *3DOC* experiments in Section 5.2. The prompt uses
Python syntax, and begins with an "import" statement that indicates functions available from the fine-grained
object localization API (Section 4.2). Users queries are converted to lower case, the ending punctuation is
removed, and prepended with a # (which represents a comment in Python). Then, the preprocessed query
is added to the few-shot prompt after an empty line. The modified prompt is then processed with an LLM.
For multi-round dialog, additional interactions or corrections are added after the response from the LLM
without an empty line separator (as demonstrated in the second and third few-shot examples).

```
from utils import find_objects, sort_objects, is_close

# find the wooden chair
def go_find_it():
  chairs =
    find_objects('chair', distractors=['sofa', 'bench', 'stool', 'desk'])
  chairs = sort_objects(chairs, attributes=['wooden'])
  return chairs[0]

# find a backpack with white polka dots on it
def go_find_it():
  backpacks
    = find_objects('backpack', distractors=['pillow', 'jacket', 'shirt'])
  backpacks = sort_objects(backpacks, attributes=['white polka dots'])
  return backpacks[0]
# no, my backpack is also yellow
def go_find_it():
  backpacks
    = find_objects('backpack', distractors=['pillow', 'jacket', 'shirt'])
  backpacks
     = sort_objects(backpacks, attributes=['white polka dots', 'yellow'])
  return backpacks[0]

# find the pants on the dresser
def go_find_it():
  pants = find_objects
    ('pants', distractors=['shirt', 'socks', 'shoes', 'dress'])
  dressers = find_objects
    ('dresser', distractors=['bookshelf', 'bed', 'desk', 'chair'])
  pants = is_close(pants, dressers)
  return pants[0]
# no, my pants are also red
def go_find_it():
  pants = find_objects
    ('pants', distractors=['shirt', 'socks', 'shoes', 'dress'])
  pants = sort_objects(pants, attributes=['red'])
  dressers = find_objects
    ('dresser', distractors=['bookshelf', 'bed', 'desk', 'chair'])
  pants = is_close(pants, dressers)
  return pants[0]

# find the apple next to the microwave
def go_find_it():
  apples = find_objects
    ('apple', distractors=['pear', 'tomato', 'orange', 'bowl'])
  microwaves = find_objects
    ('microwave', distractors=['dishwasher', 'sink', 'refrigerator',])
  apples = is_close(apples, microwaves)
  return apples[0]
```

Listing 1: Few-shot prompt used for experiments in simulation.

Table 2: *3DOC* Statistics

| | |
|---|---|
| Num of Object Categories | 10 |
| Num of Unique Objects | 72 |
| Num of Unique Intrinsic Attributes | 53 |
| Num of Unique Scenes | 100 |
| Num of Unique Scene Layouts | 500 |
| Num of Extrinsic Attribute Episodes | 50 |
| Num of Intrinsic Attribute Episodes | 1,713 |

## B  *3DOC* Statistics

Table 2 provides additional details on the composition of the *3DOC* dataset.

## C  Qualitative Examples

A video presenting qualitative examples of our *GoFind* agent executing the *FindThis* task in both simulation and in the real-world is provided in the supplemental material.

## D  Dataset and Code Release

The *3DOC* dataset and source code to reproduce the results presented in Section 5.2 will be publicly released.

