# OpenReview forum: "FindThis: Language-Driven Object Disambiguation in Indoor Environments"
_robot-learning.org/CoRL/2023/Conference — CoRL 2023 Poster_

### Official Review · Reviewer_r5uR · 2023-07-16

**Confidence:** 4
**Originality:** Very Good
**Technical Quality:** Excellent
**Clarity Of Presentation:** Good
**Impact:** 4

**Recommendation:**

Weak Accept: I recommend accepting the paper, but will not argue for my recommendation if the majority of other reviewers have a different opinion.

**Review:**

## Strengths
1. **Novel task/dataset**: As the authors point out, natural language is often ambiguous. Many current frameworks ignore this fact, and the FindThis task seeks to correct the oversight. The construction of the dataset is solid, especially the attention to distractors and the distinction made between intrinsic and extrinsic attributes.

2. **Modeling contributions**: As far as I can tell, the integration of delta vectors into the object ranker is novel. The factorization of the task into discrete steps is sensible and the way the steps (location, ranking, measurement) are implemented into an API makes the method flexible w.r.t. the parser.

3. **Strong experimental results**: Their model shows strong improvements over relevant baselines. The baselines ablate the relevant parts of the model.

4. **Real-world demonstration**: the authors include a convincing real-world demonstration of their system.


## Weaknesses
The first weakness is my only major issue with the paper; the second is more minor.


1. **"Dialogue" is a stretch**: I have concerns both about the setting and the way it is pitched. Firstly, the fact that the dialogue is triggered by a localization mistake hurts the proposed method's generalizability. This is fine for simple actions like finding an object, but what if the agent were to also have state-changing actions. In this case, waiting for the agent to make a mistake before correcting it would be untenable. A more general system would detect underspecification *before* acting and ask for clarification.

This ties into my second concern, which is with the pitch of the interaction as a dialogue. The dialogue here is really a scripted 2-stage interaction and doesn't involve language generation from the model side. A more realistic interaction might involve a multiturn dialogue that takes place over the course of the agent performing a series of actions.

2. **Hard-coded/limited agent**: There are several components in the modeling pipeline that seem hardcoded. Firstly, the exploration strategy is fixed and it is assumed that the agent has explored the environment beforehand (this is mentioned in the Limitations section). Additionally, the script followed by the localization API seems pretty much fixed and has limited breadth. As far as I can tell, program diversity is limited. From the prompt in Appendix A, it seems like there are only 2 types of programs (find_objects -> sort_objects and find_objects -> find_objects -> is_close). This makes parsing task is quite simple and means that it could be handled by intent indentification and slot-filling rather than a full-blown API. Real language, on the other hand, is likely quite a bit more complex and requires a larger set of primitives.

Another limitation that reduces the method's generalizability is the lack of complex spatial relationships and the lack of compositionality. It is assumed that there is only one relationship that objects can have (near). The authors leave other relationships for future work. Even focusing just on "near", their treatment is overly simplistic. Near is both gradable and context dependent, meaning that a) the meaning of near is vague and it isn't clear what is and isn't near; and b) "near" means different things depending on what the objects are. "Near" in "my house is near the airport" doesn't mean the same thing as "the mug is near the apple". 0.5m may be "near" on a kitchen table, but it is far if the objects are located inside a fridge.

The intrinsic and extrinsic relations could also be composed, leading to a more realistic and challenging scenario. For example, the templates could be extended to include utterances like "Find the orange mug near the large sink", which would also introduce more complex programs.


(The following is a suggestion rather than a weakness): The authors point out that certain objects have typical environments. Past work has found that this kind of commonsense information is often encoded in LLMs, so the system could include a planning stage in which the agent determines likely locations to explore.

3. **Missing related work**: The authors have related works sections for referring expressions, language-conditioned navigation, and 3D semantic mapping, but none for ambiguity and underspecification in robotics, which has been studied in the past. Specific work that also has underspecified instructions (just from a cursory Google scholar search) includes:
- https://ieeexplore.ieee.org/stamp/stamp.jsp?tp=&arnumber=9197315
- https://ieeexplore.ieee.org/stamp/stamp.jsp?tp=&arnumber=8392364
- https://dl.acm.org/doi/pdf/10.1145/3382507.3418863
- https://journals.sagepub.com/doi/full/10.1177/0278364918760992



**Quality Of The Limitations Section:**

Limitations are addressed clearly

**Questions For Rebuttal:**

(In addition to the comments in the weaknesses)

1. I would argue that there is a distinction to be drawn between ambiguity and underspecification, and that ambiguity is a particular type of underspecification, but not all underspecification is ambiguity. How do you define ambiguity, and does your definition distinguish between ambiguity and underspecification?


**Robotics Focus:**

Sufficient demonstration on hardware

**Summary Of Paper:**

The paper focuses on finding objects in physical environments when provided with underspecified language commands. The authors first introduce a new environment for testing physical object identification from language commands as well as a dataset for evaluation. It is a household environment with objects that have realistic properties and collocation statistics. They then introduce a system that attempts to tackle the task. The proposed GoFind system is a pipeline system consisting of an object detector, an object localizer, and an instruction parser.

The system is able to find objects with a greater success rate than the baseline systems. The authors also show the benefits of having an interactive setting, where a user can provide additional information to a model if it fails to find the correct object.

**Summary Of Recommendation:**

Based on the paper's strengths, I think it should be accepted. The proposed dataset and system address a gap in current research on interactions with language. Tackling ambiguity will be an important component to scaling up systems that interact with language and enabling use by real people. By framing the disambiguation process as a dialogue (at least conceptually) I think this paper is aiming in the right direction, and takes some important initial steps in that direction. My main concerns are described in the weaknesses section. With additional positioning w.r.t. past work on ambiguity the paper and more hedging around the framing of the interaction as a dialogue, I would likely have recommended a strong accept.

---

> ### Author Response · Authors · 2023-08-15
> **Response to r5uR**
>
> Thank you for your suggestions and thoughts on our work. We address your concerns below. Please let us know if any additional clarification is required.
>
> > The fact that the dialogue is triggered by a localization mistake hurts the proposed method's generalizability. This is fine for simple actions like finding an object, but what if the agent were to also have state-changing actions. In this case, waiting for the agent to make a mistake before correcting it would be untenable.
>
> We agree that irreversible, state-changing actions should be handled carefully. For example, if the agent needs to crack an egg, it should be certain that it is the correct action or check with the user. However, as the reviewer points out, such actions do not exist in the FindThis problem statement, thus fall beyond the scope of this work. We will add a note about this to the discussion section.
>
> > The model does not generate language.
>
> Correct. The model presents the user with one or more image crops of candidate objects, not language. The crops can be programmatically verified, and if the target object is not found additional rounds of dialog are triggered. If the model generated free-form language, this would require a human-in-the-loop to respond to the model’s outputs, which would preclude large-scale benchmarking as is feasible with our proposed dataset (3DOC).
>
> > Hard-coded/limited agent. exploration strategy is fixed; the localization API seems pretty much fixed and has limited breadth; Real language [...] requires a larger set of primitives.
>
> We partially agree.
>
> We agree there is scope to add API calls to better handle extrinsic relationships. As we discuss in L208-209, “Future work may explore additional API calls that capture more refined spatial relationships such as ‘left of’ or ‘behind’.”
>
> However, the existing API flexibly handles intrinsic attribute differentiation due to its natural language interface. Similarly, we find that frontier exploration adequately explores new environments to collect views of the visible objects, which is similar to the findings in related works such as [12].
>
> > The intrinsic and extrinsic relations could also be composed, leading to a more realistic and challenging scenario… would also introduce more complex programs.
>
> Agreed. In fact, we do combine intrinsic and extrinsic relations in the paper. See results in Table 1 columns 11 and 12 labeled “extrinsic + intrinsic”. This does introduce more complex programs such as `find_objects → sort_objects → (store as a), find_objects → (store as b), is_close(a, b)`.
>
> > Near is both gradable and context dependent
>
> In the 3DOC dataset, extrinsic relationships such as “near” are manually annotated in a context dependent manner -- that is, the annotator looks at the scene and determines if the “near” relationship is appropriate.
>
> In our approach we find that a set threshold for “near” performs reasonably well (e.g., the best performing method in Table 1 achieves 47.4% success on single round extrinsic relationship episodes). However, as discussed in L208-209, a more refined method is left for future work.
>
> > (The following is a suggestion rather than a weakness): The authors point out that certain objects have typical environments. Past work has found that this kind of commonsense information is often encoded in LLMs, so the system could include a planning stage in which the agent determines likely locations to explore.
>
> Thanks for this suggestion. We agree that additionally leveraging the LLM to efficiently explore the environment would be a really useful direction for the future.
>
> > I would argue that there is a distinction to be drawn between ambiguity and underspecification, and that ambiguity is a particular type of underspecification, but not all underspecification is ambiguity. How do you define ambiguity, and does your definition distinguish between ambiguity and underspecification?
>
> In this work, we define underspecification at the language+scene level, rather than at the language level alone. Consider the following example, if there is only one cup in the environment, the sentence "bring me my cup" is not underspecified, but if there are two cups it is. Our dataset does not include examples of target objects with ambiguous names -- i.e., names that have more than one meaning such as horn (the instrument) vs horn (the part of an animal). Thus, ambiguity and underspecification are used interchangeably in this work. We will add a clarification to the paper.
>
> > Additional related work.
>
> Thank you for these suggestions. We will add these to the related work section.

---

### Official Review · Reviewer_rPQj · 2023-07-19

**Confidence:** 4
**Originality:** Good
**Technical Quality:** Good
**Clarity Of Presentation:** Good
**Impact:** 3

**Recommendation:**

Weak Accept: I recommend accepting the paper, but will not argue for my recommendation if the majority of other reviewers have a different opinion.

**Review:**

The problem addressed in this paper is very interesting and relevant for service robots. Furthermore, the authors have shown the relevance of their approach in real world experiments. Nonetheless the reviewer has some concerns regarding:
- The clarity of the paper. The proposed method is somehow unclear (delta vectors) and no supporting figure is provided
- The use of templates/parsing. Although the authors provided comparisons to benefits of using a parser, the reviewer is not convinced by such an approach (cf. rebuttal questions)


**Quality Of The Limitations Section:**

Additional details required

**Questions For Rebuttal:**

1) The authors show in Table 1 results showing the superiority of parsing-based method over non-parsing ones. The authors find this comparison unfair. From the reviewer understanding, the authors only use sentences  "Find X", for which a parser would most likely be 100% accurate. With natural language, parsers would not be 100% accurate and error would accumulating in the proposed pipeline.

2) The accompanying video shows that the proposed method is able to handle free-from natural language. So the reviewer does not understand why the authors created a benchmark with only template-based sentences. Such a benchmark could be solved with rule-based approaches. There is not enough linguistic diversity.

3) Please provide either in the paper or in the annex, samples of sentences of the Go_find task which were used in experiments to demonstrate the ability to handle open-vocabulary. So far, the reviewer only find sentences "Find X  " which support the comments 1) and
2).

4) Lines 194-196, the authors mention background categories for calculating $\tau$.  This part is unclear for the reviewer.  if an LLM is used to specify similar objects to the foreground object (Line 196), why are they classified as background objects?

5) The authors proposed the delta vector vector to score each of object ROI. The reviewer finds this approach interesting instead of the usual cosine similarity. Unfortunately the description of the method is unclear and does not appear in any of the figures. E.g. Why  is the difference make only in the CLIP text embedding space? What does this difference represent?

6) Figure 3 is too small and barely readable. As such it does not provide any added value to the paper. The same comment is valid for Figure 4.

**Robotics Focus:**

Sufficient demonstration on hardware

**Summary Of Paper:**

This paper proposes the GoFind algorithm, which allows a robot to be instructed with a fetching task. The algorithm pipeline detecting ROI of objects and classify them given the parsed input sentence. The authors also propose a dataset 3DOC on which their algorithm is validated.

**Summary Of Recommendation:**

Although the task addressed by the authors is highly relevant and some interesting ideas are proposed (use of delta vectors), the following
 limitations unfortunately hinders the paper:
 *template-based sentences,
 *unfair comparison with SOA,
 *lack of sentence diversity,
Therefore reviewer cannot recommend yet this paper for CoRL.

Update:
Most of the comments have been convincingly addressed by the authors. Nonetheless, the reviewer is not fully convinced by the parsing approach given the simple template sentences. The authors implies that LLM parsers work with natural sentences but this is not shown nor evaluated in the paper. A quick review of the literature suggests that LLM are not 100% accurate:

[1] On Robustness of Prompt-based Semantic Parsing with Large Pre-trained Language Model: An Empirical Study on Codex

[2] ZEROTOP: Zero-Shot Task-Oriented Semantic Parsing using Large Language Models

[3] BenchCLAMP: A Benchmark for Evaluating Language Models on Semantic Parsing

---

> ### Author Response · Authors · 2023-08-15
> **Response to rPQj (1/2)**
>
> Thank you for your suggestions and thoughts on our work. We address your concerns below. Please let us know if any additional clarification is required.
>
> > The authors show in Table 1 results showing the superiority of parsing-based method over non-parsing ones. From the reviewer understanding, the authors only use sentences "Find X", for which a parser would most likely be 100% accurate. With natural language, parsers would not be 100% accurate and error would accumulating in the proposed pipeline.
>
> The benefits of parsing-based methods illustrated in Table 1 do depend on 100% parsing accuracy. We agree with the reviewer that a basic parser would not be 100% accurate for natural language instructions.
>
> However, for this exact reason, our proposed approach uses an LLM to parse instructions. In our real world trials with strong LLMs (e.g., GPT-3 or PALM 540B), we have not observed parsing errors. Thus, we expect little to no error will accumulate due to the parsing step.
>
> > The accompanying video shows that the proposed method is able to handle free-from natural language. [Benchmark uses] template-based sentences. There is not enough linguistic diversity.
>
> Our benchmark focuses on linguistic diversity in the intrinsic and extrinsic attributes mentioned in the instructions.
>
> In particular, the 3DOC dataset includes 53 different intrinsic attributes mentions (e.g., turquoise, striped, wicker, etc.) and 9 different extrinsic attribute phrases (e.g., on the, above the, next to the, etc.). We argue that handling such intrinsic and extrinsic attribute diversity is a first-order challenge in solving the FindThis task (L29-33). This is because intrinsic and extrinsic attributes are the primary mechanisms that people use to indicate which instance of an object they want. Thus, we make intrinsic and extrinsic attribute diversity the primary focus of the FindThis task.
>
> We use templates (Section 3.3) to generate instructions in simulation because this allows us to do systematic evaluation with controlled generation over intrinsic and extrinsic attributes. And (as the reviewer points out) we demonstrate that by using LLMs, our method generalizes to “free-form, natural language” in the real-world. With this approach to evaluation, we avoid the challenges associated with generating bias-free, large-scale natural language datasets in simulation.
>
> > Provide sample sentences of the FindThis task
>
> Thank you for the suggestion. Five multi-round samples are provided below, and we include additional examples to the Appendix.
>
> Examples:
>
> (1) R1: Find the wicker basket. R2: No, my basket is also white.
>
> (2) R1: Find the feather hat. R2: No, my hat is also green.
>
> (3) R1: Find the green towel. R2: No, my towel is also striped.
>
> (4) R1: Find the white plate. R2: No, my plate is also porcelain.
>
> (5) R1: Find the black hat. R2: No, my hat is also woven.
>
> > Lines 194-196, the authors mention background categories for calculating. This part is unclear for the reviewer. if an LLM is used to specify similar objects to the foreground object (Line 196), why are they classified as background objects?
>
> To recap, background categories are used to set a similarity threshold (L193-194). In other words, detections that are more similar to a background category than the target, foreground category are discarded.
>
> While the space of background categories is large, only the subset of “nearest neighbor” categories are required to set the threshold because other categories will have an extremely low similarity.  Through few-shot prompting, we use an LLM to generate the subset of “nearest neighbors.”
>
> Specifically, these categories should be visually similar but not synonyms for the target category. For example, for the target object “apple” we might want the background categories to include “pear,” “tomato,” and “orange.” We will add these details to the paper.

---

> > ### Author Response · Authors · 2023-08-15
> > **Response to rPQj (2/2)**
> >
> > > The reviewer finds [delta vectors] interesting instead of the usual cosine similarity. Unfortunately the description of the method is unclear and does not appear in any of the figures. E.g. Why is the difference make only in the CLIP text embedding space? What does this difference represent?
> >
> > Thank you for pointing this out. We provide additional details below. We will add these details along with a supporting figure to the paper.
> >
> > The difference is taken in the CLIP text embedding space because points in this space (such as the CLIP text embedding of “mug”) notionally represent the canonical (or most common) view of an object (such the most common image of a mug). This is a result of the contrastive learning objective used in CLIP.
> >
> > We take the difference between (a) the name of the object (e.g., “mug”) and (b) the attribute-object phrase (e.g., “orange mug”). This difference is a vector from the canonical view of the object to the canonical view of the object when it has a specific attribute. Thus, this vector represents the attribute of interest (e.g., “orange”). When we project the CLIP image embedding of detected objects onto this vector, we get a measure of how much of the attribute is seen in the detections.
> >
> > > Figures 3 and 4 are too small.
> >
> > We will increase the size of these figures. Thank you for the suggestion.

---

### Official Review · Reviewer_pqhb · 2023-07-20

**Confidence:** 5
**Originality:** Good
**Technical Quality:** Very Good
**Clarity Of Presentation:** Good
**Impact:** 3

**Recommendation:**

Weak Accept: I recommend accepting the paper, but will not argue for my recommendation if the majority of other reviewers have a different opinion.

**Review:**

The research topic is a very interesting application of VLM and LLM to real-world tasks in robotics. A unique feature of this research is the disambiguation based on multiple dialogs with the user. Much of the content of this paper has been compressed, and some details seem to have been omitted. I present my questions and concerns below.

The specificity, open-vocabulary, and interactive mentioned by the author are all important for mobile service robots.
However, open-vocabulary can be solved by simply using LLM. It seems to me that this is a problem that has already been solved. The proposed method does not seem to solve open-vocabulary in a new way. Thus, there were no big surprises for me.

The proposed method is able to deal with instruction ambiguity and wrong requests by Multi-around interaction.
However, having multiple linguistic interactions in every instruction may be burdensome for the user.
Could the proposed method use the previous interactions if the same ambiguous instruction is repeated the next time as the first?
In other words, it does not seem to allow the memory and knowledge exchanged once to be used the next time.
Some action or consideration of the above may be needed.

**Quality Of The Limitations Section:**

Additional details required

**Questions For Rebuttal:**

How did the robot acquire a map of its environment? In the proposed method, the robot seems to search for unexplored areas of the environment using the Frontier approach. However, I did not find any description of how the map was created in the experimental section. Were the maps created by SLAM? The Frontier approach should be applied to unexplored areas of the map. Therefore, it was my understanding that the map was constructed during exploration. The author may be able to clarify on the above.

The author wrote as if he had compared NLMap and CoW in an experiment. However, I could not find any results for these methods. In fact, it seems that some open-vocabulary object detectors and with/without parsing were compared, not with these methods.
To avoid this confusion, the authors could clarify their description of Baselines.

The format of the citation of the paper is not consistent.
Authors need to closely check and correct all citations with respect to their descriptions.
Also, there are too many arXiv preprints in the citations. To some extent, this is unavoidable due to the nature of this industry. However, arXiv is not peer-reviewed and can undermine the credibility of a paper. Therefore, I encourage authors to make an effort to reduce arxiv preprint citations as much as possible.



What is "background categories"? Details on this are unclear.

Figure 1.
It is difficult to tell what the image sequence and arrows indicate.
Some of what the figure shows is unclear.

The success rate is defined as SR in section 3.2 and is also redefined in section 5.2.

**Robotics Focus:**

Sufficient demonstration on hardware

**Summary Of Paper:**

This paper proposed a method for robots to search for the location of specific objects using vision and language models.
The proposed method is divided into two phases: a phase in which objects in the environment are investigated by search using an open vocabulary object detector, and a phase in which the robot's execution command is output using LLM from human commands.
In the experiments, several open vocabulary object detectors were compared using the Habitat-Matterport3D dataset.


**Summary Of Recommendation:**

This paper incorporates the latest machine learning techniques into an important robotics problem. The focus of the problem is very interesting.
The content seems appropriate for this conference.
However, perhaps due to the page limit of the conference, the richness of the content is compressed and some details are omitted.
In addition, more detailed and extensive experiments and comparisons are required to demonstrate the proposed method.

---

> ### Author Response · Authors · 2023-08-15
> **Response to pqhb (1/2)**
>
> Thank you for reviewing our work and sharing your thoughts. We address your concerns below. Please let us know if any additional clarification is required.
>
> > The specificity, open-vocabulary, and interactive mentioned by the author are all important for mobile service robots. However, open-vocabulary can be solved by simply using LLM. It seems to me that this is a problem that has already been solved.
>
> Not quite. LLMs only address the open-vocabulary problem in the text domain. Whereas the FindThis task goes beyond text by requiring grounded visual understanding.
>
> Specifically, this work focuses on object disambiguation using intrinsic and extrinsic attributes -- all of which can be specified using an open-vocabulary. For example, consider our real-world "bowl discovery" experiment (in the supplemental video) in which the human indicates that their bowl is "full of cereal." Because we consider an open-vocabulary setting, the human could have said their bowl was filled with any number of things (e.g., fruits, candies, or marbles) and the robot would need to properly ground the query in the visual domain to respond appropriately.
>
> Our results in Table 1 indicate that FindThis is far from a solved problem. For example, with 1 round of interaction the best performing method only achieves a 52% success rate when the agent can present up to 5 candidates to the user. Similarly, with 2 rounds of interaction the best method has a success rate of 63.2%.
>
> Furthermore, Table 1 shows that our proposed approach, which introduces the use of CLIP delta vectors for intrinsic attribute differentiation, helps but does not solve this new benchmark either. Specifically, as discussed in L278-282: “we find that our approach of using CLIP delta vectors for intrinsic attribute differentiation consistently improves top-1 SR (row 2 vs. 3, 5 vs. 6, and 8 vs. 9). For example, with the ViLD+CLIP representation intrinsic top-1 SR improves by +1.6 points for k=1 (38.7 → 40.3) and +1.4 points k=2 (45.0 → 46.4) (row 5 vs. 6). With ViLD representations, we observe larger improvements in top-1 SR of +4.0 points (23.8 → 27.8) for k=1 and +5.4 points (29.6 → 35.0) for k=2 (row 2 vs. 3).”
>
> > How did the robot acquire a map of its environment?
>
> The robot is given an occupancy map delineating navigable regions in the environment, and then uses Frontier exploration [43] to collect views of the environment.
>
> Specifically, the occupancy map can be created using standard mapping techniques (e.g., SLAM) in real world settings. As discussed in L225-228, “the real-world occupancy map only delineates walls and other immovable structures (e.g., a kitchen island) but does not include movable objects such as tables or chairs. The robot uses RGB-D and lidar sensors to update the occupancy map in real-time, which allows navigating to waypoints without collisions.”
>
> Frontier exploration [43], is used to collect views of the scene. As discussed in L230-233, “at each timestep, visible regions within a fixed radius of the agent are marked as ‘explored.’ Then, the agent selects the nearest waypoint along the boundary (or frontier) of the explored area to navigate to next. The agent updates the explored regions along the way and stops exploring when all of the navigable regions have been marked ‘explored.’”
>
> > I could not find any results for [NLMap and CoW]. To avoid confusion, the authors could clarify their description of Baselines.
>
> We reimplemented NLMap [38] and CoW [12], with minor variations required for addressing the FindThis task (such as multi-round interaction). We report results for these reimplementations in Table 1 rows 1 and 4 for NLMap [38] and row 7 for CoW [12]. We will clarify the descriptions of these baselines and add the details to the Appendix.
>
> > What is "background categories"? Details on this are unclear.
>
> To recap, background categories are used to set a similarity threshold (L193-194). In other words, detections that are more similar to a background category than the target, foreground category are discarded.
>
> While the space of background categories is large, only the subset of “nearest neighbor” categories are required to set the threshold because other categories will have an extremely low similarity. Through few-shot prompting, we use an LLM to generate the subset of “nearest neighbors.”
>
> Specifically, these categories should be visually similar but not synonyms for the target category. For example, for the target object “apple” we might want the background categories to include “pear,” “tomato,” and “orange.” We will add these details to the paper.

---

> > ### Author Response · Authors · 2023-08-15
> > **Response to pqhb (2/2)**
> >
> > > Could the proposed method use the previous interactions if the same ambiguous instruction is repeated the next time as the first? In other words, it does not seem to allow the memory and knowledge exchanged once to be used the next time. Some action or consideration of the above may be needed.
> >
> > Great question! The reviewer is correct, we only consider the episodic version of the task where there is no memory of previous interactions. However, we agree that in certain real world deployments a user may want the robot to remember, for example, that their laptop has stickers on it. That said, this long-term deployment version of the task falls beyond the scope of this work. And we leave it to future studies to appropriately define that variation of the FindThis task.
> >
> > > The format of the citation of the paper is not consistent; Figure 1 is unclear
> >
> > Thank you for pointing this out. We will fix the citations and update Figure 1 by annotating the meaning of each component.

---

### Official Review · Reviewer_qZX4 · 2023-07-21

**Confidence:** 4
**Originality:** Very Good
**Technical Quality:** Very Good
**Clarity Of Presentation:** Very Good
**Impact:** 3

**Recommendation:**

Weak Accept: I recommend accepting the paper, but will not argue for my recommendation if the majority of other reviewers have a different opinion.

**Review:**

Strengths
- I like the task and interactive human-centric setting proposed and the formulation provided for the FindThis task is sensible
- Prompting an LLM for distractor objects seems like a good way to calibrate similarity outputs from CLIP embeddings.
- The open-source is constructed to include diverse objects and valid distractor objects in each scene

Weaknesses
- Real experiments don't have as much complexity as simulated, for example one round only has 2 objects in it (laptop with/without sticker), so the agent will always succeed within 2 rounds. Similarly, there are only 3 mugs so a randomly performing agent would also achieve quite a high success rate.
- Success being counted as >10% of the cropped image containing the object seems like a weird metric, it seems to me the metric should also include the % of the target object included in the crop, with some high threshold required for success (say >80%). Otherwise, a crop could be clearly centered on an adjacent object and only include part of the target object, but still be counted as correct. Does this ever happen in the experiments provided? Why is such a low threshold for performance chosen for the success metric?
- The 3D representation used in GoFind includes lots of redundant views of objects and no way to associate them as the same object.
- This work implicitly assumes that every query describes an object that actually exists in the scene, since the robot has no mechanism for responding "I can't find it", which is a natural response to a query. This should be discussed in the limitations and ideally mentioned in the problem statement, or if not evaluated in experiments. Additionally, having examples in the 3DOC dataset where queries do not exist would be helpful as a benchmark.

Comments
- In the 3 considerations presented, I feel "natural language" would be a better term than "open vocabulary", which sometimes has some connotations of fixing a set of supported terms before system deployment, such as how "open vocabulary" object datasets still have a restrictive set of object descriptions. On page 2 line 44, "open-vocabulary, natural language" is a bit redundant in my opinion, wouldn't any true natural language system be inherently open vocab?
- The conversation is described as a "multimodal dialogue" (page 4 line 112), which I feel is a misleading personification since the human agent is only able to respond with text, and the LLM only with an image. It is a dialogue which includes multimodal elements, but isn't a full 2-way multimodal dialogue.
- I feel there should be a short section in RW about using code generation from LLMs as an interface for an agent's actions, for example ViperGPT, Code as Policies, and ConceptFusion to name a few. I'm not sure of related work on human-in-the-loop interactive task completion with an LLM, but I'd be surprised if there wasn't (ChatGPT!)
- There are some additional citations on 3D semantic/language maps that could make the section more complete: ConceptFusion, CLIP-Fields, LERF, Distilled Feature Fields, Panoptic Lifting.
- The design decision of using CLIP delta vectors for object attributes is interesting, but more motivation for this method should be provided in the description. Also, the main text describes which lines in the table represent the delta ablation, but the table only refers to "basic parsing". I feel the table could be made clearer, possibly by changing "+ ours" to "+ delta"
- page 6 line 205, "an language query"-> "a language query"
- A discussion of failures should be included given some of the ~50% success rates in simulated scenes.

**Quality Of The Limitations Section:**

Limitations are addressed clearly

**Questions For Rebuttal:**

Questions
- Why is there such an imbalance between intrinsic and extrinsic properties in the dataset?
- What are the specific 10 categories, and 72 objects? images in the appendix or an exhaustive list would be very helpful
- Only 2 rounds are considered, why not more?

**Robotics Focus:**

Sufficient demonstration on hardware

**Summary Of Paper:**

This paper proposes a task of interactive language-based object search in a navigation environment. To that end, it defines an alternating dialogue where a human sends progressively more specific queries and an agent replies in candidate pictures. The paper introduces a dataset which includes scanned scenes with objects placed inside, chosen to include multiple objects from the same category with potentially distracting properties. It then proposes a method for addressing this task where a 3D cached set of object detections from an open-vocab vision-language detection model is processed with a large language model to rank the candidates based on parsed attributes.

**Summary Of Recommendation:**

I think the method and dataset are thorough and contributes a useful dataset for a potentially fruitful direction of human-in-the-loop object localization. Despite some concerns and questions I've listed above, my overall rating is weak accept.

---

> ### Author Response · Authors · 2023-08-15
> **Response to qZX4 (1/2)**
>
> Thank you for your thoughtful review. We found the suggestions very useful and will incorporate them into the work. We address your concerns below. Please let us know if any additional clarification is required.
>
> > Real experiments don't have as much complexity as simulated
>
> We generally agree. We believe this highlights the advantages of simulation, where we are able to systematically increase the complexity to create the challenging 3DOC benchmark.
>
> Having said that, there is still substantial complexity in both the scene and language used in the real setting.
>
> The scene is a real office kitchen that was in active use during the experiments (note the people moving through the area in the supplemental video). Thus, there are several cups, bowls, water bottles, kitchen equipment, etc. that could be confused for the targets. Furthermore, we selected challenging attributes such as (laptop) “with stickers on it”, “upside down” (mug), and (bowl) “full of cereal” to increase complexity.
>
> > Success being counted as >10% of the cropped image containing the object seems like a weird metric; [Use a higher threshold] Otherwise, a crop could be clearly centered on an adjacent object and only include part of the target object, but still be counted as correct.
>
> We selected 10% based on initial experiments with a variety of objects.
>
> Recall, the metric is calculated as the percentage of the crop that contains the ground truth segmentation mask of the target object.
>
> With lower thresholds we did observe the failure mode pointed out by the reviewer, where crops centered on other background objects were counted as correct because they included a few pixels of the target. By increasing the threshold to 10%, this issue was resolved.
>
> On the other hand, with high thresholds we run into an opposite problem: crops clearly centered on objects get marked as unsuccessful. For example, consider a “pencil case” (shaped as a narrow cylinder) that gets detected at a viewing angle such that it goes diagonally through the image crop. In such situations, a large percentage of the crop will be background, and the crop would be marked as unsuccessful. Accordingly, we selected a lower threshold to avoid this alternative issue.
>
> We will add this discussion to the appendix, including visual examples.
>
> > The 3D representation used in GoFind includes lots of redundant views of objects and no way to associate them as the same object.
>
> Agreed. Our 3D representation includes multiple views of each object because in certain circumstances some attributes can only be seen from some angles and not others. For example, to identify if an open laptop has stickers on the case, an agent would need to see the laptop from the back.
>
> That said, redundant views of an object can easily be removed from the candidate set as a post-processing step before the candidates are presented to the user. Specifically, the approach for de-duplicating views in [38], which uses 3D spatial proximity, could be adapted for this task in future work.
>
> > This work implicitly assumes that every query describes an object that actually exists in the scene, since the robot has no mechanism for responding "I can't find it", which is a natural response to a query. This should be discussed in the limitations and ideally mentioned in the problem statement, or if not evaluated in experiments.
>
> Thank you, this is a great suggestion! We will discuss this in the limitations and mention it in the problem statement as suggested by the reviewer.
>
> > Why is there such an imbalance between intrinsic and extrinsic properties in the dataset?
>
> The imbalance is a result of how the two splits are annotated.
>
> The intrinsic properties require annotations at the object level. After annotation, the objects can be procedurally placed in environments to generate a large number of evaluation episodes. In short, annotating 10s of objects can produce 100s of episodes.
>
> In contrast, extrinsic relationships depend on the specific placement of the objects in the environment and are not procedurally generated. Thus, 1 annotation results in 1 episode.
>
> > What are the specific 10 categories, and 72 objects? images in the appendix or an exhaustive list would be very helpful
>
> The 10 categories are: plate, pencil case, tape, bowl, lunch box, basket, hat, towel, mug, and toy. We will add images and an exhaustive list to the appendix. The details will also be included in the open-sourced code.
>
> > Only 2 rounds are considered, why not more?
>
> Two rounds were considered because the objects in our dataset (e.g., mugs or towels) typically only have two primary intrinsic attributes (e.g., color and pattern). Note that our proposed approach can easily handle additional rounds of dialog.

---

> > ### Author Response · Authors · 2023-08-15
> > **Response to qZX4 (2/2)**
> >
> > > "natural language" vs. "open vocabulary"; “multimodal dialogue”; RW on code generation 3D semantic/language maps; typos; discussion of failure modes; design of CLIP delta vectors
> >
> > Thank you for all of these suggestions! We will clarify that we truly mean “natural language” without any restrictions. We will also clarify that the human and agent use different modalities for communication. Additionally, we will expand the related work section as suggested, fix the typos, add a discussion of failure modes, and expand the description of CLIP delta vectors in the text and provide more motivation.

---

### Author Response · Authors · 2023-08-15
**Response to all reviewers**

We would like to thank all of the reviewers for their time providing thoughtful reviews with several great suggestions that will strengthen the paper.

Overall, we are happy that the reviewers “like the task and interactive human-centric setting” (qZX4), finding it “very interesting and relevant for service robots” (rPQj), “novel” (r5uR), and appreciate the “attention to distractors and the distinction made between intrinsic and extrinsic attributes” (r5uR). Furthermore, they appreciated the proposed approach for “prompting an LLM for distractor objects” (qZX4) and found the “integration of delta vectors into the object ranker... novel” (r5uR). Finally, the reviewers liked the “strong experimental results” (r5uR), “improvements over relevant baselines” (r5uR), and “relevance of [the] approach in real world experiments” (rPQj).

We respond to specific concerns raised by the reviewers below.

---

### Decision · Program_Chairs · 2023-08-30

**Decision:**

Accept (Poster)

**Comment:**

The reviewers appreciate the novelty and formulation of the approach and the interactive human-centric setting, but questioned the experimental setup and evaluation measures, and ask for further discussion of the limitations of the approach.
After the rebuttal and author discussion phase, the reviewers agree that this paper can be accepted for CoRL. The reviewer suggestions should be thoroughly addressed in the camera-ready version of the paper.